# Statistical Analysis of SARS-CoV-2 Using Wastewater-Based Data of Stockholm, Sweden

**DOI:** 10.3390/ijerph20054181

**Published:** 2023-02-26

**Authors:** Aashlesha Chekkala, Merve Atasoy, Cecilia Williams, Zeynep Cetecioglu

**Affiliations:** 1Department of Chemical Engineering, KTH Royal Institute of Technology, 10044 Stockholm, Sweden; 2UNLOCK, Wageningen University & Research and Technical University Delft, 6708PB Wageningen, The Netherlands; 3Science for Life Laboratory, Department of Protein Science, KTH Royal Institute of Technology, 17121 Solna, Sweden; 4Department of Industrial Biotechnology, KTH Royal Institute of Technology, AlbaNova University Center, 11421 Stockholm, Sweden

**Keywords:** wastewater-based epidemiology, SARS-CoV-2, PMMoV, statistical analysis

## Abstract

An approach based on wastewater epidemiology can be used to monitor the COVID-19 pandemic by assessing the gene copy number of SARS-CoV-2 in wastewater. In the present study, we statistically analyzed such data from six inlets of three wastewater treatment plants, covering six regions of Stockholm, Sweden, collected over an approximate year period (week 16 of 2020 to week 22 of 2021). SARS-CoV-2 gene copy number and population-based biomarker PMMoV, as well as clinical data, such as the number of positive cases, intensive care unit numbers, and deaths, were analyzed statistically using correlations and principal component analysis (PCA). Despite the population differences, the PCA for the Stockholm dataset showed that the case numbers are well grouped across wastewater treatment plants. Furthermore, when considering the data from the whole of Stockholm, the wastewater characteristics (flow rate m^3^/day, PMMoV Ct value, and SARS-CoV gene copy number) were significantly correlated with the public health agency’s report of SARS-CoV-2 infection rates (0.419 to 0.95, *p*-value < 0.01). However, while the PCA results showed that the case numbers for each wastewater treatment plant were well grouped concerning PC1 (37.3%) and PC2 (19.67%), the results from the correlation analysis for the individual wastewater treatment plants showed varied trends. SARS-CoV-2 fluctuations can be accurately predicted through statistical analyses of wastewater-based epidemiology, as demonstrated in this study.

## 1. Introduction

The COVID-19 pandemic, which is caused by the contagious severe acute respiratory syndrome coronavirus 2 (SARS-CoV-2), has accounted for almost 645 million cases and over 6 million deaths as of November 2022 [1]. As the number of asymptomatic infected people increases, clinical survey methods are insufficient for estimating infection spread, making wastewater-based epidemiology (WBE) an attractive alternative. [2]. WBE can provide surveillance and real-time monitoring of SARS-CoV-2 transmission and help trigger pandemic responses within the community [3].

In 2019, the first COVID-19 patient-reported gastrointestinal symptoms appeared, where the stool and respiratory specimens were found to be positive for SARS-CoV-2 when analyzed by real-time reverse transcription-polymerase chain reaction (RT-PCR) [4]. SARS-CoV-2 viral particles are shed into the bodily excreta, including saliva, sputum, and feces, which are further disposed into the sewage streams [5]. A study assessed patterns of SARS-CoV-2 disease and viral load from different samples (nasopharyngeal and blood, urine, and stool samples), and reported that two out of the five patients in the study had SARS-CoV-2 detected in the stool sample [6]. Therefore, WBE can be used as an early prediction tool for the COVID-19 pandemic. Continuous monitoring of wastewater also allows for the early detection of variants and other virus mutations present in the community. The Netherlands was the first country to report the detection of SARS-CoV-2 in wastewater collected from seven different cities within the country [3]. Finland [7], Australia [8], Germany [9], Sweden [10], the Netherlands [11], and the USA [12] have all carried out early studies to successfully monitor viral signals to complement the existing public health metric.

Statistical analysis is an important process that impacts the ability to extract useful and valid information from experimental data. The application of statistics in wastewater-based epidemiology studies has received wide recognition [13,14] and includes common tests such as *t*-test, Gaussian distribution, correlation studies, and regression models, to better understand the experimental data by reducing the errors.

Experimental data as a function of case-control, cross-section, time-series, or even longitudinal studies are common study designs in epidemiology. To analyze such types of data, it is not always clear how to form correlation structures, therefore different statistical methods are used to provide the tools which can be applied to data with variability and uncertainty. Table 1 shows the different statistical methods which have been used by researchers to form a valid conclusion for WBE studies.

Many of the statistical methods listed in Table 1 have been used to establish an early wastewater-based warning system for COVID-19, by identifying the relationships between gene copy number of SARS-CoV-2 and wastewater characterization [27,28,29,30]. For instance, Dai et al. (2022) successfully applied a novel Bayesian statistical framework on the basis of functional principal component analysis, to forecast viral concentrations in wastewater [27]. Feng et al. (2021) used various statistical methods (i.e., nested ANOVA, Tukey’s post hoc test, Spearman’s rank correlation) to establish a link between COVID-19-related clinical data and gene copy number of SARS-CoV-2 in wastewater [28]. Therefore, we here applied different statistical methods to understand how the gene copy number of SARS-CoV-2 in wastewater corresponded with wastewater characterization and COVID-19-related clinical data in Stockholm, Sweden.

In this study, the data from the wastewater treatment plants (WWTP) and weekly clinical data from Stockholm were analyzed to understand the relationship between wastewater parameters and clinical data obtained during the first year of the pandemic [31]. In detail, the weekly data of wastewater (i.e., flowrate, biomarker virus PMMoV, the SARS-CoV-2 gene copy number), spiked bovine virus, and clinical data were used for the statistical analyses. We collected wastewater from six inlets of three WWTPs in Stockholm, Sweden: Henriksdal, Bromma, and Käppala WWTPs, representing six different regions that cover almost all of Stockholm.

## 2. Methodology

The methodology of the study includes data gathering and statistical analyses, as briefly illustrated in Figure 1. 

### 2.1. Data Gathering

The experimental data were obtained from the research group at the KTH Royal Institute of Technology, Sweden, and are available as open source via DataCentre (https://covid19dataportal.se/). The data from the first sampling protocol of Perez-Zabaleta et al. [26] were used. Detailed information regarding sampling, sample preparation, viral concentration, RNA extraction, RT-qPCR analysis, and calculations, is available in Jafferali et al. [10] and Perez-Zabaleta et al. [31].

Briefly, 318 wastewater samples were collected from six inlets of three different WWTPs: Three inlets from the Bromma WWTP named Hässelby (59.3662° N, 17.8600° E), Riksby (59.3316° N, 18.0657° E), and Järva (59.3818° N, 17.9932° E); two inlets from Henriksdal WWTP named Sickla (59.3071° N, 18.1199° E) and Henriksdal (59.3123° N, 18.1080° E); and one inlet from Käppala WWTP (59.3529° N, 18.2183° E), between April 2020 (week 16) to June 2021 (week 22). The WWTP:s serve a varied population size: Bromma WWTP (Stockholm Vatten och Avfall) treats wastewater from approximately 377,500 inhabitants, Henriksdal WWTP (Stockholm Vatten och Avfall) 862,100 inhabitants, and Käppala WWTP (Käppala Association) 700,000 inhabitants. 

At the WWTPs, an equal volume (50 mL from each of the inlets) of flow-proportional composite samples (taken before any biological or chemical treatment) was collected each day for one week and stored at 4 °C. 350 mL of raw wastewater from each inlet of each WWTP was transported to the laboratory on ice weekly. Once the wastewater samples arrived at the laboratories, the samples were kept at 4 °C until concentration and RNA extraction, which was usually performed the same or the next day [31]. 

The wastewater samples (10 mL per analysis) were concentrated through double filtration by using 10 kDa cut-off centrifugal ultrafilters (Sartorius), as previously described by Jafferali et al. [10]. Moreover, 20 μL of bovine coronavirus (BCoV) was spiked into 50 mL of wastewater sample before filtration as an external reference. The calculation and explanation of how BCoV was used as an external reference are provided by Jafferali et al. [10]. RNA extraction and concentration were performed by using the miRNeasy Mini Kit (Qiagen, Chatsworth, CA, USA).

The reverse transcriptase quantitative polymerase chain reaction (RT-qPCR) was performed as previously described by Jafferali et al. [10] for the quantification of SARS-CoV-2, BCoV, and pepper mild mottle virus (PMMoV) on each sample, in duplicate. PMMoV was used for data normalization and, in this manuscript, as the PMMoV factor. The weekly N-gene copy number per inlet was adjusted for the variations in PMMoV levels per week (PMMoV factor) for each inlet: Henriksdal, Sickla, Hässelby, Järva, Riksby and Käppala, using Equation (1).


(1)
$$\begin{array}{c} N\;\mathrm{or}\;P(\mathrm{gene}\;\mathrm{copy}\;\mathrm{number}/\mathrm{week})=\\ \;\;\;\;\;\;\;\;\;\;\;\;C(\mathrm{gene}\;\mathrm{copy}\;\mathrm{number}/\mathrm{mL})\times (\mathrm{total}\;\mathrm{flowrate}\;\mathrm{per}\;\mathrm{week}\;(\mathrm{mL}/\mathrm{week})) \end{array}$$



(2)
$$\mathrm{PMMoV}\;\mathrm{Factor}=\frac{P_{x,y}(\mathrm{PMMoV}\;\mathrm{gene}\;\mathrm{copy}\;\mathrm{number}/\mathrm{week})}{{\mathrm{Averange}\;P(\mathrm{PMMoV}-\;\mathrm{gene}\;\mathrm{copy}\;\mathrm{number}\;\mathrm{per}\;\mathrm{week})}_{\left(\mathrm{Inlet}\;y\right)}}$$


*C* represents the gene copy number per mL of wastewater, which was obtained by converting the Ct values of either the N-gene or PMMoV gene to gene copy numbers per reaction with the standard curves. The copy numbers per reaction were then recalculated to gene copy number per mL of wastewater, by correcting for the respective dilutions of input RNA to PCR reaction (4:10 N-gene or 1:10 PMMoV), RNA elution volume, and initial wastewater sample volume (55 μL RNA extracted from 10 mL wastewater). The volume of wastewater per week in each inlet (sample points) was obtained by multiplying the flow rates from the WWTPs in m^3^/day, converted to mL, by 7 days. 


(3)
$$N_{\mathrm{corrected}}=\frac{N_{\left(\mathrm{week}\;x,\mathrm{Inlet}\;y\right)}}{{\mathrm{PMMoV}\;\mathrm{Factor}}_{\left(\mathrm{week}\;x,\mathrm{Inlet}\;y\right)}}$$


N values were corrected for each WWTP by summing the N_corrected_ values from the corresponding inlets, by applying the PMMoV factor as is shown in Equation (3). For instance, the summation of the N_corrected_ from Hässelby, Järva, and Riksby inlets provided the total N-gene copy number per week for the entire Bromma WWTP. Similarly, Henriksdal WWTP used the values from the Henriksdal inlet and Sickla inlet, whereas Käppala WWTP has only one inlet. The N_corrected_ values from Bromma, Henriksdal, and Käppala WWTPs were summed to provide the total N-gene copy number per week for the full Stockholm area. 

The clinical data, including the number of COVID-19 positive cases, intensive care unit (ICU) numbers, and deaths following infection with SARS-CoV-2, were obtained from the Swedish Public Health Agency (https://experience.arcgis.com/experience/19fc7e3f61ec4e86af178fe2275029c5 accessed on 1 July 2021).

### 2.2. Statistical Analysis

#### 2.2.1. Descriptive Statistics (Box Plots)

Descriptive statistics are a common tool used to organize and describe characteristics or factors of a given sample set of data points. These methods help to describe the midpoint of the data sets and the spread of scores, dispersion, or variance.

The descriptive statistics are used to graphically plot various parameters (flowrate m^3^/day, PMMoV Ct value, gene copy number per WWTP per week with bovine factor) for the three WWTPs: Henriksdal, Bromma, and Käppala. The box plots are used to graphically represent these parameters, as this allows the representation of five of the most common features of the data sets: minimum and maximum range values, lower and upper quartiles, and the median. These plots can further be used to provide a more straightforward way to compare the datasets based on the features [32]. Descriptive analysis was carried out using the Statistical Package for the Social Sciences (IBM SPSS 26).

#### 2.2.2. Principal Component Analysis (PCA)

The experimental data presented in this study consists of a combination of time-series and case-controlled data sets. To interpret such complex data sets, PCA is a statistical tool used to drastically reduce the dimensionality in an interpretable way while also preserving most of the information in the data [33]. Initially, the data sets are dimensionally reduced to fit on a two-dimensional plot, which would allow for understanding how the various quality parameters are related to each other and capture the variance within the datasets. This test was performed on the data sets for the total Stockholm, as well as the individual WWTPs (Henriksdal, Bromma, and Käppala). The plots for the PCA were obtained by Origin, IBM SPSS, and Excel.

#### 2.2.3. Correlation Analysis

Correlation analysis is a statistical tool that is used to test the correlation between quantitative variables. Performing this analysis allows us to form predictions on the future behavior of the dataset based on the relationship exhibited by the variables [34]. Correlation analysis was carried out for the Stockholm data set and the data set present for the individual WWTPs: Henriksdal, Bromma, and Käppala. The correlation analysis was carried out using Pearson’s correlations as an indicator of the strength of linear relationships within the dataset. Correlation is significant at the 0.01 level, considering a 2-tailed test. The correlation analysis was performed in IBM SPSS and tabulated in Excel.

## 3. Results and Discussions

The experimental data analyzed in this study used RT-PCR to detect and quantify the presence of SARS-CoV-2 RNA from wastewater samples collected from week 16 of 2020 to week 22 of 2021, from three different WWTPs (Henriksdal, Bromma, and Käppala) in Stockholm. The distribution analyses for the variables, flow rate (m^3^/day), average Ct value of PMMoV, and the gene copy number/WWTP per week, normalized with bovine factor [10] in each of the regions, are shown in Figure 2. The average flow rate of the raw wastewater samples was 98,650 m^3^/day in the Henriksdal WWTP, 107,249 m^3^/day in the Bromma WWTP, and 89,534 m^3^/day in the Käppala WWTP (Figure 2a). The average Ct value of PMMoV detection was 26.90 (Henriksdal WWTP), 27.66 (Bromma WWTP), and 24.49 (Käppala WWTP) (Figure 2b). Finally, the average value of the SARS-CoV-2 N-gene copy number per week per WWTP, adjusted by the bovine factor, was 2.71 × 10^18^ (Henriksdal), 6.85 × 10^17^ (Bromma), and 2.77 × 10^18^ (Käppala) (Figure 2c).

### 3.1. Relation between SARS-CoV-2 Cases and Water Parameters Considered in This Study

To understand the relationship between the influent wastewater characteristics and the SARS-CoV-2 positive cases in the individual regions of Stockholm and the total Stockholm datasets, a principal component analysis was carried out. The PCA plot shows a comprehensive picture of the interactions between the SARS-CoV-2 viral loads and the flow rate of wastewater (m^3^/day). The data set presented from week 16 of 2020 to week 22 of 2021 was subjected to PCA and projected on a 2D domain, with two main PCs as the component axis, to represent a converted correlation (or a lack of it) among the variables. Figure 3 presents the relation/interaction of the wastewater parameters with the positive cases of the SARS-CoV-2-infected patients for individual regions in Stockholm.

PCA analysis was carried out on the parameters for the data sets in the individual analysis to group the data. The two principal components (PC1 and PC2) describe the variations present in the data sets and account for varied loadings of the characteristic parameters and can indicate clusters of the samples based on their similarity. From Figure 3, it is inferred that the parameter case number for the individual regions is well grouped concerning PC1 (37.3 %) and PC2 (19.67%). This means that even though there is a population difference between Henriksdal, Bromma, and Käppala wastewater treatment plants, the case numbers grouped well. However, for the other parameters, PMMoV Ct value, flowrate (m^3^/day), and gene copy number/WWTP (per week with the bovine factor), there is continuous flexibility in the variables rather than a ‘tightly clustered’ flexibility, which also means that these parameters are independent.

It should be noted that when plotting the data on a 2D plot, the PCs provide the directions that summarize the variance of the data set and not a structural pathway with biological meaning. Through this analysis, we can conclude that the data present from the sewage water samples fluctuate randomly rather than in a continuous manner. A previous study performed PCA on a similar WBE application of SARS-CoV-2 and found a modest association between PMMoV CT, per capita flow rate, and per capita designed flow rate [35] (Figure 3). Served population, flow rate, and designed flow rate made negligible contributions to PC2 and PC3, indicating a very weak association with PMMoV Ct.

A loading plot is a part of the PCA and is used to show how strongly each characteristic influences a principal component [36]. The angles between the parameter vectors provide further details on how parameters can be correlated with each other. Figure 4 presents the loading plot for the total Stockholm data. From this plot, it can be inferred that the positive case numbers, hospitalized patient numbers or intensive care unit (ICU) numbers, and deaths following infection with SARS-CoV-2, were positively correlated. The data on the case numbers, ICU numbers, and deaths were obtained from the Swedish Public Health Agency based on the population reported. A positive correlation between these parameters signifies that they are a function of each other. However, the flow rate of the influent wastewater samples and gene copy number of SARS-CoV-2 per week for each WWTP, with the bovine factor, was negatively correlated. As shown in Figure 4, these two parameters are signifying that the gene copy number of SARS-CoV-2 found in the wastewater streams, when normalized with the bovine factor, is not a function of the flow rates of the wastewater samples from the WWTP. To further confirm the correlation between the variables, correlation analysis was carried out.

### 3.2. Correlation Analysis

WBE can function as a tool to surveil SARS-CoV-2 spread in the community. This is because wastewater detection is a key parameter that forms a relationship with the case numbers of the SARS-CoV-2-infected population. Influent wastewater parameters, such as flow rate (m^3^/day), or levels of PMMoV (Ct value) and SARS-CoV-2 (gene copy number with bovine factor per WWTP), can be analyzed for correlation with data on case numbers, ICU numbers, and deaths follwoing SARS-CoV-2 infection, as provided by the public health authority. Thus, the relationship between the parameters can be examined. The correlation analysis was performed here for the datasets considering the entire Stockholm area (Figure 5) and for the individual regions of Stockholm: Henriksdal (Figure 6a), Bromma (Figure 6b), and Käppala (Figure 6c).

Figure 5 represents the whole Stockholm area from week 16 of 2020 to week 22 of 2021. Statistically significant correlations were observed between wastewater characteristics and the SARS-CoV-2 clinical data (0.419 to 0.95 *p*-value < 0.01). The positively correlated parameters are highlighted in the green boxes (high correlation coefficient) and yellow boxes (low correlation coefficient). The negatively correlated parameters are highlighted in the red boxes.

As presented in Figure 5, the gene copy number was positively correlated with PMMoV Ct values (0.954 *p*-values < 0.01), ICU numbers (0.749 *p*-values < 0.01) and case numbers (0.711 *p*-values < 0.01) for Stockholm.

Based on Figure 6a, the case number is positively correlated with PMMoV Ct values (0.582, *p*-value < 0.01). The correlation analysis of Henriksdal data showed that the case number is a function of the PMMoV biomarker and the gene copy number of SARS-CoV-2.

Similar results were observed in correlation analysis of the Bromma data (Figure 6b): PMMoV Ct values (0.37, *p*-value < 0.01) and gene copy number of SARS-CoV-2 (0.657, *p*-value < 0.01) were positively correlated with the case number. It can be concluded that the case number of SARS-CoV-2 in Bromma is a function of the PMMoV biomarker and the gene copy number of SARS-CoV-2/WWTP.

Figure 6c represents correlation analysis for the Käppala data. There is a negative correlation between the flow rate (m^3^/day) and case number (−0.491, *p*-value < 0.01), as well as the flow rate (m^3^/day) and PMMoV Ct value (correlation coefficient: −0.286). However, the case number is positively correlated with the PMMoV Ct value (0.498, *p*-value <0.01) and gene copy number of SARS-CoV-2 (0.465, *p*-value < 0.01). Based on the results, we can infer that the case number of the SARS-CoV-2 in Käppala is a function of the PMMoV biomarker and the gene copy number of SARS-CoV-2. Moreover, the flow rate (m^3^/day) is not a function of case numbers and the biomarker PMMoV in Käppala.

## 4. Limitations

WBE is an effective tool, and it can transform the wastewater infrastructure that can be extrapolated to provide a public health outlook. The epidemiological data that were used in this study can be further improved. The data were varied, as the auxiliary information needed, such as the system boundary of the catchment, and the exact population served by each treatment facility is not precise.

Further, the experimental methods in this study produced a greater variability, indicating scope for optimization in detection and quantification methodology. Environmental factors (e.g., wastewater temperature, pH, suspended solids) and the methods or chemicals used for wastewater treatment have a significant effect on the detection methods. The analysis conducted in this study does not account for other factors, such as the percentage of working individuals, the variant of the SARS-CoV-2 and the seasons for when the influent samples were collected, which are key contributors to the correlation study and need to be taken into consideration.

Moreover, the RT-PCR results used here did not include information on the SARS-CoV-2 variants, which may impact secretion into the wastewater, and thereby the level of correlation to, for example, positive case numbers. Further, understanding the virus transmission or degradation of viral RNA between different environmental compartments is an important aspect of the detection and quantification process.

Generating valid statistical results and interpretations from the present study depends on the underlying assumptions made during the study. To further extrapolate the results from the study, statistical methods need to be developed to suitably analyze the data using standardized protocols. Therefore, it is not easy to form a comparative study, as the parameters and characteristics of the statistical tools selected in other studies are not the same, resulting in different conclusions from the statistical tests.

## 5. Future Recommendations and Conclusions

Statistical analysis of SARS-CoV-2 viral RNA monitoring for approximately 55 weeks, in wastewater samples from three different WWTPs of Stockholm, was performed here. The results unravelled a significant correlation with case numbers and other clinical data for the SARS-CoV-2-infected population, and support the applicability of WBE, to provide surveillance of COVID-19. However, currently, there is no standardized method for the detection of SARS-CoV-2 viral loads in wastewater systems, which poses a problem when trying to derive conclusions for global wastewater systems.

Results obtained from WBE can be used to detect and manage this, and possibly future pandemics, more effectively. However, the lack of consistency in the wastewater parameters, which are used as part of WBE, must be addressed in future studies. This could include standardizing the sample collection and processing strategies, methods to validate the sensitivity and detection limits for viral RNA concentration, and biomarkers used as part of the study. Homogeneous protocols for WBE would allow for global comparisons and assessment of the results, which would further allow for collaborations of databases targeted in one place.

In our study, the municipal wastewater samples from three regions of Stockholm were tested and statistically analyzed to validate the results from the experimental methodology. The statistical tests were performed to understand the interactions between wastewater parameters and the clinical data (case numbers, death, and ICU numbers) for the individual regions and Stockholm as a whole. In light of our findings:Samples were collected from three different WWTPs (Henriksdal, Bromma and Käppala), which serve six different regions in Stockholm, and were analyzed for PMMoV levels (Ct value) and SARS-CoV-2 gene copy number/WWTP per week with consideration given to the bovine factor. Based on the statistical distribution of the obtained data and the flow rate (m^3^/day) for each WWTP, the difference in the dataset might be related to the capacity of the WWTP.By examining the PCA plot and loading plot for Stockholm, it is evident that the data from the wastewater samples exhibit random fluctuations instead of a continuous pattern. These fluctuations could be attributed to variances in the wastewater itself, the population that the WWTP serves, or the presence of various strains of SARS-CoV-2 in circulation.Upon correlating the parameters for Stockholm, a statistically significant positive correlation was observed between the wastewater characteristics and the available clinical data on SARS-CoV-2, with correlation coefficients ranging from 0.42 to 0.95. Nonetheless, the correlations were found to differ when conducting the analysis for specific regions.

## Figures and Tables

**Figure 1 ijerph-20-04181-f001:**
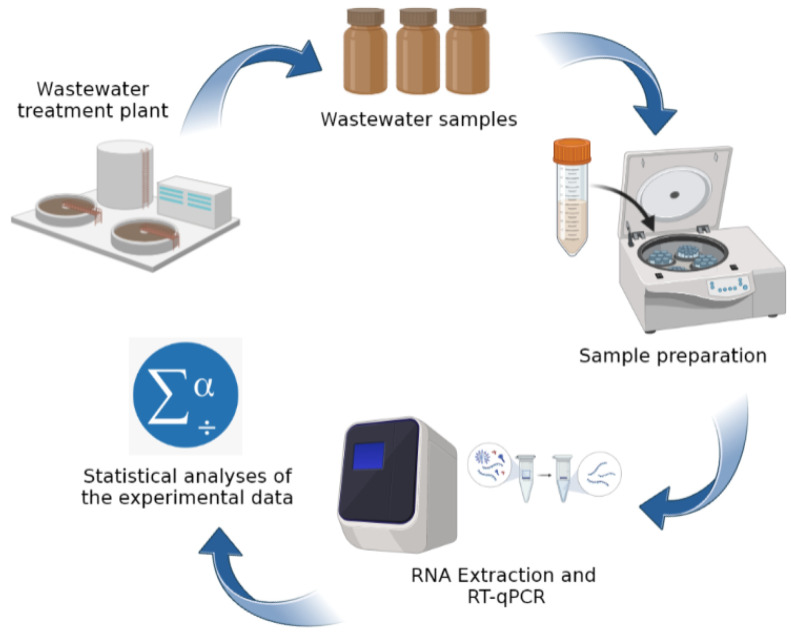
Schematic of the experimental procedure (created through BioRender.com).

**Figure 2 ijerph-20-04181-f002:**
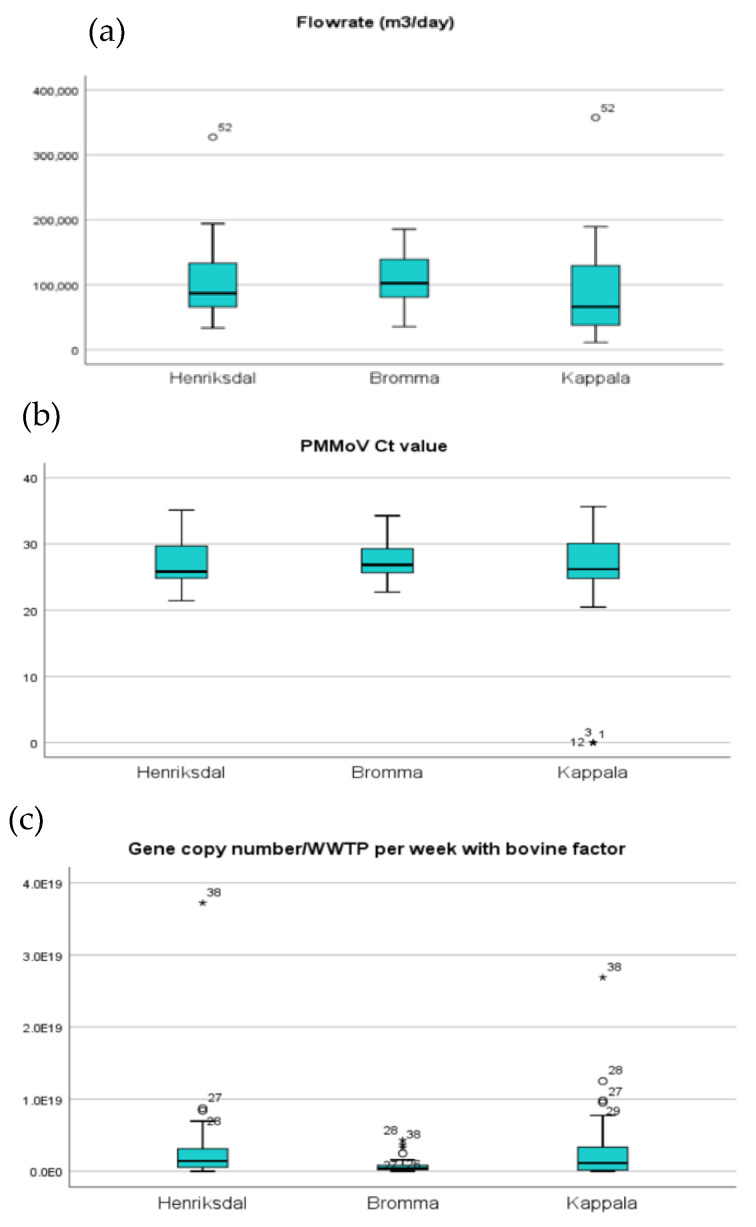
Distribution of the parameters for the individual regions in Stockholm during the study period. (**a**) Flowrate (m^3^/day). (**b**) PMMoV levels (Ct value). (**c**) SARS-CoV-2 levels (N-gene copy number per week per WWTP, adjusted by the bovine factor).

**Figure 3 ijerph-20-04181-f003:**
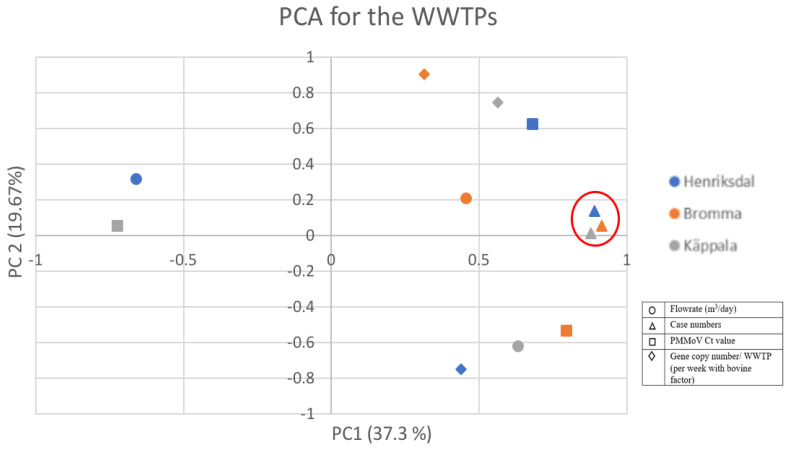
PCA plot illustrating similarity/dissimilarity between wastewater parameters and the SARS-CoV-2-positive cases of different regions in Stockholm.

**Figure 4 ijerph-20-04181-f004:**
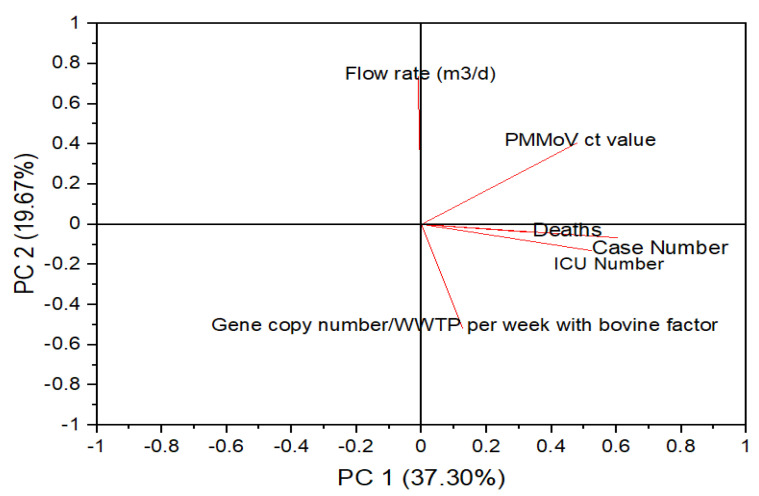
PCA loading plot, illustrating the relation and interaction among the wastewater parameters and the SARS-CoV-2-positive cases, ICU number, and deaths in Stockholm.

**Figure 5 ijerph-20-04181-f005:**
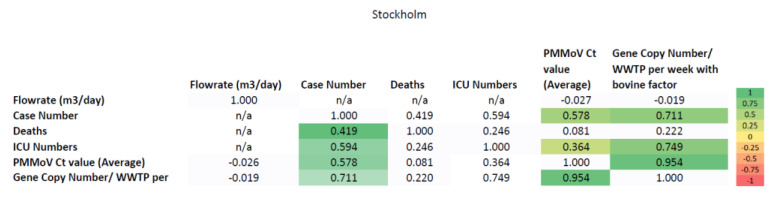
Correlation analysis between PMMoV Ct value and gene copy number/WWTP per week, with the bovine factor, for Stockholm. The positively correlated parameters are highlighted in green (high correlation coefficient) and yellow (low correlation coefficient). The negatively correlated parameters are highlighted in the red boxes (n/a: not applicable).

**Figure 6 ijerph-20-04181-f006:**
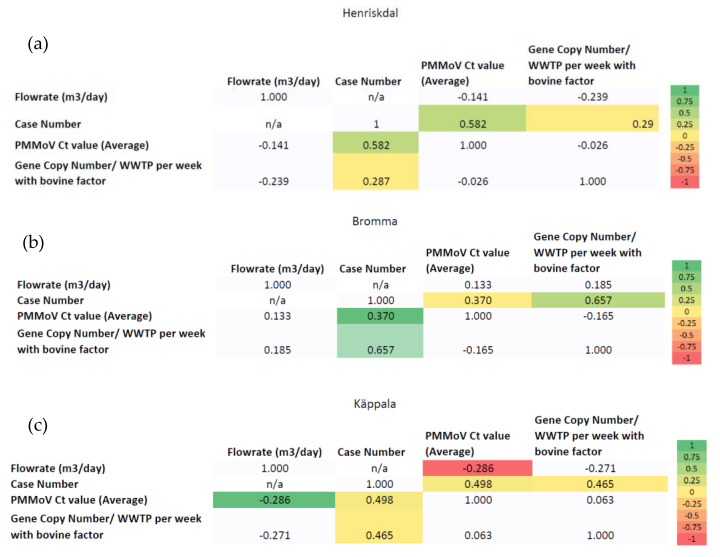
Correlation analysis for the individual regions (**a**): Henriksdal WWTP; (**b**): Bromma WWTP and (**c**): Käppala WWTP (n/a: not applicable).

**Table 1 ijerph-20-04181-t001:** Review of different statistical methods for wastewater-based epidemiology studies.

Method	Significance of Method	Reference
ANOVA and *t*-TEST	These methods can be applied to test the significance of differences between the two means. Significance level refers to the likelihood that the random variable chosen is not representative of the population. The lower the significance level, the more confident you can be in replicating your data.	[15,16]
Gaussian Distribution	The data should follow an exponential rise in each of the parameters so that the data follows a bell-shaped curve. The area under the curve plotted between “gene copy no./ wastewater treatment plant (WWTP) and weeks” will tell us how effective the method is in quantifying the viral loads. A comparative Gaussian graph can be formed to validate the method used.	[17,18]
ARIMA Models	These models are time-series models, which are used to reveal a reliable and meaningful statistical model that can be used for future analysis. They are instrumental for modelling the temporal dependency structure of time-series data, especially for series that have a cyclic or repeating pattern, given that the data changes with trends, periodic changes, and other random distortions. This model is used for fitting the time series data for hepatitis, influenza, and even SARS-CoV-2.	[19,20]
Monte Carlo Method	A probabilistic model to assess the uncertainty of a parameter, for example in wastewater analysis. The most effective quantification method for uncertainty and variability is to assign a probability density function to each parameter. This method allows us to perform sensitivity analysis, which will represent the % of influence of each experimental parameter of the outcome.	[21]
Regression Analysis (using GAM and LOESS)	A method to explore which independent parameter has a significant effect on the outcome.	[22]
Functional Distribution Analysis (FDA)	A statistical method was specifically developed to analyze temporal data.	[23]
Functional Principal Component Analysis (FPCA)	Used to analyze temporal patterns. The patterns can help us understand the extent of the accuracy of the outcome or the accuracy of the experimental data.	[23]
FANOVA	A suggested way to analyze the association between the functional data (outcome) and the co-variates.	[23]
Fisher’s Exact Test	Used to form a correlation matrix between each of the parameters selected to the presence of a positive or negative correlation.	[24,25]
Generalized Additive Model for Location, Scale and Shape (GAMLSS)	This is a regression model selected if the data do not follow a Gaussian distribution.	[26]

## Data Availability

The data are obtained from DataCentre (https://covid19dataportal.se/).

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
