# Peer review of "Statistical Analysis of SARS-CoV-2 Using Wastewater-Based Data of Stockholm, Sweden"

_ijerph, 2023, doi:10.3390/ijerph20054181_

Round 1

Reviewer 1 Report

The wastewater analysis for the SARS-CoV-2 has always been an important study after the onset of the pandemic. Manuscript is well written and objectives and methodology are detailed and well explained. I will recommend acceptance after minor revisions.

My minor suggestions are below.

In the methodology, please mention the number of wastewater samples, sampling frequency, sampling period, and mode of sampling. Also include the population served by wastewater treatment plants.

Authors may also include USA in line number 49.

Author Response

Manuscript Title: Statistical analysis of SARS-CoV-2 wastewater-based data of Stockholm  Article Type: Original research paper

Comments and Responses

General Response

We very much appreciate the very constructive comments and suggestions of the reviewers. These comments lead to interesting discussions and have been given serious consideration and we have revised the manuscript accordingly.

Reviewer #1

The wastewater analysis for the SARS-CoV-2 has always been an important study after the onset of the pandemic. Manuscript is well written and objectives and methodology are detailed and well explained. I will recommend acceptance after minor revisions.

Comment 1.1

In the methodology, please mention the number of wastewater samples, sampling frequency, sampling period, and mode of sampling. Also, include the population served by wastewater treatment plants

Response 1.1

Thank you for your comment. The following paragraph has been added to the manuscript:

The wastewater of Stockholm was monitored between April 2020 (week 16) to June 2021 (week 22). Overall 318 samples were collected from six influents of three different WWTPs, which are serving a varied population size: Bromma WWTP (Stockholm Vatten och Avfall) which treats wastewater from approximately 377,500 inhabitants, Henriksdal WWTP (Stockholm Vatten och Avfall) with 862,100 inhabitants, and Käppala WWTP (Käppala Association) with 700,000 inhabitants. Approximately 500 mL of raw wastewater (taken before any biological or chemical treatment) from each WWTP were transported to the laboratory on ice weekly.

Upon receiving the samples, an equal volume of flow-proportional-composite samples was collected each day for one week and stored at 4 °C before transferring to the laboratory. Once the wastewater samples arrived at the laboratories, the samples were kept at 4 °C until concentration and RNA extraction, which was usually performed the same or the next day [31]. The reverse transcriptase quantitative polymerase chain reaction (RT-qPCR) was conducted on each sample in duplicate after RNA extraction.

Comment 1.2

Authors may also include USA in line number 49.

Response 1.2

A study analyzing the wastewater samples for SARS-CoV-2 surveillance in the USA is now added.

Reviewer 2 Report

The current manuscript entitled “Statistical analysis of SARS-CoV-2 wastewater-based data of Stockholm” by Chekkala et al. deals with SARS-CoV-2 from six inlets of three wastewater treatment plants covering six regions of Stockholm, Sweden. After a careful reading, I found this manuscript interesting and suitable for publication in IJERPH after minor revisions. My specific comments are:

1.      Add the country name in the title. Also, add “using” after SARS-CoV-2.

2.      The abstract lacks major numerical findings. Also, units are not correctly written (line 24: 3 should be in superscript).

3.      Line 34: be specific and mention the month/year.

4.      Line 44: change “X” to a multiplication symbol.

5.      Be consistent while using “&” and “and”.

6.      Eq. 1: correct multiplication symbol.

7.      Improve the figure quality and units.

8.      Correlation figures: correct the decimal points to be uniform. Some have none, some ranging from 1-3.

9.      Correct all typographic and syntax errors present in the manuscript.

Author Response

Reviewer #2

The current manuscript entitled “Statistical analysis of SARS-CoV-2 wastewater-based data of Stockholm” by Chekkala et al. deals with SARS-CoV-2 from six inlets of three wastewater treatment plants covering six regions of Stockholm, Sweden. After a careful reading, I found this manuscript interesting and suitable for publication in IJERPH after minor revisions. My specific comments are:

Comment 2.1

Add the country name in the title. Also, add “using” after SARS-CoV-2.

Response 2.1

The title of the manuscript has been updated given below:

“Statistical analysis of SARS-CoV-2 using wastewater-based data of Stockholm, Sweden”

Comment 2.2

The abstract lacks major numerical findings. Also, units are not correctly written (line 24: 3 should be in superscript).

Response 2.2

The following parts have been included in the abstract:

“Furthermore, when considering the data from the whole of Stockholm, the wastewater characteristics (flow rate m3/day, PMMoV Ct value, and SARS-CoV gene copy number) were significantly correlated with the public health agency's report of SARS-CoV-2 infection rates (0.419 to 0.95, p-value <0.01). However, the PCA results showed that the case numbers for each wastewater treatment plant were well grouped concerning PC1 (37.3%) and PC2 (19.67%). The results from the correlation analysis for the individual wastewater treatment plants (WWTP) showed varied trends.”

Comment 2.3

Line 34: be specific and mention the month/year.

Response 2.3

The month/year has been added.

Comment 2.4

Line 44: change “X” to a multiplication symbol.

Response 2.4

Based on Comment 4.3, this part has been deleted.

Comment 2.5

Be consistent while using “&” and “and”.

Response 2.5

The “&” are replaced with “and”

Comment 2.6

Eq. 1: correct multiplication symbol.

Response 2.6

Eq. 1 is revised.

Comment 2.7

Improve the figure quality and units.

Response 2.7

The figures have been updated.

Comment 2.8

Correlation figures: correct the decimal points to be uniform. Some have none, some ranging from 1-3.

Response 2.8

The figures for the correlation analysis have been updated accordingly.

Comment 2.9

Correct all typographic and syntax errors present in the manuscript.

Response 2.9

The manuscript is now revised.

Reviewer 3 Report

The manuscript by Chekkala et al. is an article on “Statistical analysis of SARS-CoV-2 wastewater-based data of Stockholm”. It is well written, well organized, and provides interesting information. I appreciate the efforts for visualizing data in an excellent pictoral way. My evaluation of this work is very positive. Still, I suggest some minor edits that could further improve the quality of the manuscript, and this manuscript should be accepted after minor revision.

1, Add the graphical abstract highlighting important article points.
2, Line 24. “m3/day” should be superscript and revised in the whole manuscript.   
3, Line 77. Write the full name of WWTP the first time.   
4, Line 96-98. Please provide the coordinates of each sampling site. 

Author Response

Reviewer #3

The manuscript by Chekkala et al. is an article on “Statistical analysis of SARS-CoV-2 wastewater-based data of Stockholm”. It is well written, well organized, and provides interesting information. I appreciate the efforts for visualizing data in an excellent pictoral way. My evaluation of this work is very positive. Still, I suggest some minor edits that could further improve the quality of the manuscript, and this manuscript should be accepted after minor revision.

Comment 3.1

Add the graphical abstract highlighting important article points.

Response 3.1

The graphical abstract has been included as given below:

Comment 3.2

Line 24. “m3/day” should be superscript and revised in the whole manuscript.    

Response 3.2

“m3/day” is changed to m3/day in the whole manuscript.

Comment 3.3

Line 77. Write the full name of WWTP the first time.    

Response 3.3

Line 88 has been updated as follows:

In this study, the data from the wastewater treatment plants (WWTP)…”

Comment 3.4

Line 96-98. Please provide the coordinates of each sampling site. 

Response 3.4

The following paragraph has been added to the manuscript:

“Briefly, the wastewater samples were collected from overall six inlets of three WWTPS: three inlets of the Bromma WWTP -Hässelby (59.3662° N, 17.8600° E), Riksby (59.3316° N, 18.0657° E) and Järva (59.3818° N, 17.9932° E), two inlets from Henriksdal WWTP - Sickla (59.3071° N, 18.1199° E) and Henriksdal (59.3123° N, 18.1080° E), and one inlet from Käppala WWTP (59.3529° N, 18.2183° E).”

Author Response

Reviewer #4

Wastewater-based epidemiology is a useful tool to monitor the development of the COVID-19 pandemic. To correlate the amount of SARS-CoV-2 viral genome, PMMoV genomes in wastewater to different clinical data in several statistical analyses would be interesting to many readers. However, major and minor revisions are needed:

Comment 4.1

Line 27-28, the conclusion of this study is “SARS-CoV-2 fluctuations can be accurately predicted through statistical analyses of wastewater-based epidemiology, as demonstrated in this study”. How accurate of the prediction of SARS-CoV-2 fluctuations? In the results, many indicators for whole Stockholm and individual regions had a different pattern, such as the correlation analysis. Could these statistical analyses differences affect the predication?

Response 4.1

Thank you for pointing out this issue. Indeed, the prediction of the fluctuations depends on several factors. Therefore, the sentence has been revised as below:

“SARS-CoV-2 fluctuations can be predicted through statistical analyses of wastewater-based epidemiology, as demonstrated in this study.”

Comment 4.2

Line 39-41. Wrong citation. The citation in the text is a review, the origin citation should be: Holshue, Michelle L., et al. "First case of 2019 novel coronavirus in the United States." New England journal of medicine (2020).

Response 4.2

Thank you for pointing it out. The reference has been revised.

Comment 4.3

Line 43-44. “on average, 6.3 – 1.8 x 108 SARS-CoV-2 RNA gene copies per gram of faeces are found”, in the cited reference, it wrote 6.3x105– 1.8 x 108 , but the same problem, this citation is not origin one. The data in original publication is “viral load as high as 6·8 log10 copies per g of stool for patient 4 and 8·1 log10 copies per g of stool for patient 5”, which is about 6.3x106 and 1.3x108 . It is debatable to draw the conclusion of the range of viral genome copies in stool just based on 2 patients. Provide other references or rewrite the sentence.

Response 4.3

The sentence has been rewritten as below:

“In 2019, the first COVID-19 patient-reported gastrointestinal symptoms, where the stool and respiratory specimens were found to be positive for SARS-CoV-2 when analyzed by real-time reverse transcription–polymerase chain reaction (RT-PCR) [4]. SARS-CoV-2 viral particles are shed into the bodily excreta, including saliva, sputum, and faeces, which are further disposed into the sewage streams [5].  A study assessed patterns of SARS-CoV-2 disease and viral load from different samples (nasopharyngeal and blood, urine, and stool samples), and reported that two out of the five patients in the study detected SARS-CoV-2 in the stool sample [6].”

Comment 4.4

Line 49. Citation 9 and 11 are not available to access, please have a correction. In addition, it is better to use an origin scientific paper instead of a CNN report.

Response 4.4

References 9 and 11 have been replaced with scientific papers.

Comment 4.5

In table 1. Some words are in bold, such as “significant different, bell-shaped curve”. Do they have special meaning?

Response 4.5

The words were in bold to specify what each method can be used for. But it’s removed now.

Comment 4.6

Line 69 and 71. Strange citations. Citations 32-35 appeared before citation 26? Probably some errors when cited.

Response 4.6

The citations have been rearranged chronologically now.

Comment 4.7

Line 99-101. The description is confusing. The first is ‘The wastewater samples (10 mL) were concentrated’, and later, ‘bovine coronavirus (BCoV) was spiked in 50 ml of each wastewater sample’. What exactly 10 or 50 ml used for the concentration?

Response 4.7

The paragraph has been revised based on the comment:

“The bovine coronavirus (BCoV) (20 μL) was spiked into 50 ml of each wastewater sample before filtration as an external reference. The wastewater samples (10 mL) were concentrated through double filtration by using 10 kDa cut-off centrifugal ultrafilters (Sartorius) as previously described by Jafferali et al. [10]. The concentrated part was used for RNA extraction by using miRNeasy Mini Kit (Qiagen, Chatsworth, CA).”

Comment 4.8

Line 111, Eq 1. It is better to replace “flow rate” with “total flow per week”.

Response 4.8

Eq.1 is revised now.

Comment 4.9

Line 113, Eq 2. How the average gene copy of PMMoV per week calculated? Please elaborate it in Methods.

Response 4.9

In light of the fact that the data was not calculated in the present manuscript, the following references are provided for the detailed data generation and calculation:

“Jafferali et al. [10] and Perez-Zabaleta et al. [26] provide details of the calculation of PMMoV gene copy numbers.”

Comment 4.10

Line 136-137. The link is wrong, and it is not Swedish public health agency. It is arcgis website.

Response 4.10

The reference has been updated according to the comment.

Comment 4.11

Line 144. “flowrate m3/day”. Please check spacing between words and superscript in the whole article.

Response 4.11

The term “flowrate m3/day” is now revised.

Comment 4.12

Line 145. “gene copy number per WWTP per week with bovine factor”. In the Eq2, Eq3 and Methods, the gene copy was corrected by PMMoV factor. From here and following part, it seems that it is corrected by bovine coronavirus. Please explain the differences.

Response 4.12

Thank you for your comment. As a result of filtration and extraction processes, the bovine factor was used as an external reference. However, the PMMoV factor was used to normalize the data. The following sentences are included in the manuscript to clarify the difference between PMMoV and bovine factors:

“The bovine coronavirus (BCoV) (20 μL) was spiked into 50 ml of each wastewater sample before filtration as an external reference.

N values were corrected for each WWTP by summing the Ncorrected values from the corresponding inlets by applying the PMMoV factor as is shown in Eq.(3).”

Comment 4.13

Line 182-185. Wrong order for Figure 2b and 2c. And what do the labelled numbers mean in Figure 2? For example, 52 labelled on 2a, 12, 3, 1 labelled on 2c.

Response 4.13

The lines have been revised and the figures are arranged in order.

Comment 4.14

Line 207-209. “This means that even though there is a population difference between Henriksdal, Bromma and Käppala, the case numbers are the same.” This is one of the conclusions of this study, it is better to support this statement with more data. For example, what is the number of SARSCoV-2 cases in these 3 regions? Are they the same number?

Response 4.14

Due to the fact that the PCA graph does not represent a specific data point, the statement is unlikely to be supported by the case number data. To clarify the statement, the sentence has been updated as follows:

This means that even though there is a population difference between Henriksdal, Bromma and Käppala wastewater treatment plants, the case numbers grouped well.”

Comment 4.15

Line 209. From Table 2c, the PMMoV Ct value varied a lot. In Kappala, the PMMoV Ct seems ranged from 20 to more than 35. Do you have an explanation of the huge variation? In ref 30, the Ct value of PMMoV is 27.08±3.06, and in some studies, the variant is even smaller, such as Ct variance = 1.18 in D'Aoust, Patrick M., et al. "Quantitative analysis of SARS-CoV-2 RNA from wastewater solids in communities with low COVID-19 incidence and prevalence." Water research 188 (2021): 116560. The huge variant of PMMoV Ct value could lead to the conclusion that this variable is independent

Response 4.15

Thank you for the comment. The data represents almost a year of sampling points, therefore, it is very likely that several parameters (e.g. water temperature, nutrient levels, pH, microbial community, the concentration of antiviral compounds, etc.) affect both PMMoV and SARS-CoV-2 concentrations and therefore their Ct values. Nonetheless, there are no solid references/results to support the statement that PMMoV Ct values are independent.

Comment 4.16

Line 218-222. These five lines are exactly the same as the reference, without a change of single word. Similar citation also from Line 39-41. This kind of citation is not appropriate, and suggest to rewrite these sentences.

Response 4.16

The lines have been revised.

Comment 4.17

In Figure 4, there are five red lines, but with six indicators. Does one red line missing?

Response 4.17

The sixth indicator is the flowrate. The red line is almost on the 90o axis. Also, the indicator for the case number and death almost align.

Comment 4.18

Line 249-251, Figure 6, 7, and 8 should be Figure 6a, 6b, 6c

Response 4.18

Thank you for pointing it out. The figures have been changed.

Comment 4.19

Results and Discussions are combined in this article. However, most of this part is results. A poor discussion is presented. For example, the correlation analysis results showed that whole Stockholm and individual regions had different outcomes, how to explain the differences? Also, Figure 6a showed that “the flow rate (m3/day) is negatively correlated with the case number (- 0.425, p-value <0.01)”. This does not make sense. Flow rate in WWTP is affected by precipitation and snow melting, otherwise it is stable during a short period, how could the flow rate negatively correlated with the SAR-CoV-2 case number? And this negative correlation only found in 2 regions, but not in whole Stockholm. Suggest to have a thorough discussion.

Response 4.19

Thank you for the detailed comment. The results and discussion part has been revised. A correlation between flow rate and case number (as well as ICU and death number) cannot be considered. All the figures and the manuscript is revised accordingly.

Comment 4.20

Line 319, should be “six” regions instead of “three”?

Response 4.20

The related paragraph has been updated accordingly as below:

“Samples were collected from 3 different WWTPs (Henriksdal, Bromma and Käppala), which serve 6 different regions in Stockholm and analysed for PMMoV levels (Ct value) and SARS-CoV-2 gene copy number/WWTP per week with consideration given to the bovine factor. Based on the statistical distribution of the obtained data and the flow rate (m3/day) for each WWTP, the difference in the dataset might be related to the capacity of the WWTP.”

Comment 4.21

Line 327-328. Please elaborate the conclusion “The variation in these datasets may contribute to differences in the operations of the WWTP systems”. For example, the variations of SARS-CoV2 genomes in wastewater reflect its spread in society, and the samples used in this work is from influent wastewater, which is not treated in WWTP. How do the operations of each WWTP matters? The same question from conclusion 2, will “WWTPs did not follow a standardized protocol of operation” lead to wastewater sample data fluctuations?

Response 4.21

“WWTPs did not follow a standardized protocol of operation”, has been revised for better understanding as provided below:

“Samples were collected from 3 different WWTPs (Henriksdal, Bromma and Käppala), which serve 6 different regions in Stockholm and analysed for PMMoV levels (Ct value) and SARS-CoV-2 gene copy number/WWTP per week with consideration given to the bovine factor. Based on the statistical distribution of the obtained data and the flow rate (m3/day) for each WWTP, the difference in the dataset might be related to the capacity of the WWTP.

By examining the PCA plot and loading plot for Stockholm, it is evident that the data from the wastewater samples exhibit random fluctuations instead of a continuous pattern. These fluctuations could be attributed to variances in the wastewater itself, the population that the WWTP serves, or the presence of various strains of SARS-CoV-2 in circulation.

Upon correlating the parameters for Stockholm, a statistically significant positive correlation was observed between the wastewater characteristics and the available clinical data on SARS-CoV-2, with correlation coefficients ranging from 0.42 to 0.95. Nonetheless, the correlations were found to differ when conducting the analysis for specific regions.”

Comment 4.22

Please check the reference lists. A. There are several different formats in reference lists, such as ref 4 and ref 5. B. Reference 7 is duplicate. C. Strange symbol in ref 13, etc.

Response 4.22

The reference list has been revised.

Round 2

Reviewer 4 Report

The reviewed version has improved the quality of the manuscript, and the authors responded most of the concerns and questions. A few still remaining are:

1. Line 91-104. This part is newly added. Is this the description of figure 1, or should it have a separate title.

2. Line 99-100. "Upon receiving the samples, an equal volume of flow-proportional-composite samples was collected" In first paragraph, 500 ml raw wastewater was collected and transferred to lab, here an equal volume of sample was collected, what is this equal volume sample? 

3. Line 156-157. Still the wrong link of Swedish public health agency.

4. Line 359. m3/day

5. Following question 4.12 in first version. Suggest to add how the bovine factor was used to correct the gene coyp of SARS-CoV-2 in Methods.

Author Response

[IJERPH] Manuscript ID: ijerph-2184668

Manuscript Title: Statistical analysis of SARS-CoV-2 using wastewater-based data of Stockholm, Sweden

Article Type: Original research paper

Comments and Responses (Round 2)

Reviewer #4
The reviewed version has improved the quality of the manuscript, and the authors responded most of the concerns and questions. A few still remaining are:

Comment 4.1

Line 91-104. This part is newly added. Is this the description of figure 1, or should it have a separate title.

Response 4.1

Thank you for your comment. This part is integrated into the 2.1 Data Gathering section as given below:

“Briefly, the wastewater samples were collected from overall six inlets of three WWTPS: three inlets of the Bromma WWTP -Hässelby (59.3662° N, 17.8600° E), Riksby (59.3316° N, 18.0657° E) and Järva (59.3818° N, 17.9932° E), two inlets from Henriksdal WWTP - Sickla (59.3071° N, 18.1199° E) and Henriksdal (59.3123° N, 18.1080° E), and one inlet from Käppala WWTP (59.3529° N, 18.2183° E) between April 2020 (week 16) to June 2021 (week 22). Overall, 318 samples were collected from six influents of three different WWTPs, which serves a varied population size: Bromma WWTP (Stockholm Vatten och Avfall) which treats wastewater from approximately 377,500 inhabitants, Henriksdal WWTP (Stockholm Vatten och Avfall) with 862,100 inhabitants, and Käppala WWTP (Käppala Association) with 700,000 inhabitants. 

Approximately 350 mL of raw wastewater (taken before any biological or chemical treatment) from each inlet of each WWTP was transported to the laboratory on ice weekly. At the WWTPs, an equal volume (50 ml from each of the inlets) of flow-proportional-composite samples was collected each day for one week and stored at 4 °C before transferring them to the laboratory. Once the wastewater samples arrived at the laboratories, the samples were kept at 4 °C until concentration and RNA extraction, which was usually performed the same or the next day [31]. Prior to RNA extraction, daily samples from each inlet were mixed equally volume-wise per week.”

Comment 4.2

Line 99-100. "Upon receiving the samples, an equal volume of flow-proportional-composite samples was collected" In first paragraph, 500 ml raw wastewater was collected and transferred to lab, here an equal volume of sample was collected, what is this equal volume sample? 

Response 4.2

To prevent the confusion, this part is updated as given below:

“Approximately 350 mL of raw wastewater (taken before any biological or chemical treatment) from each inlet of each WWTP was transported to the laboratory on ice weekly. At the WWTPs, an equal volume (50 ml from each of the inlets) of flow-proportional-composite samples was collected each day for one week and stored at 4 °C before transferring them to the laboratory. ”

Comment 4.3

Line 156-157. Still the wrong link of Swedish public health agency.

Response 4.3

Thank you for pointing it out. The link has been updated. (https://experience.arcgis.com/experience/19fc7e3f61ec4e86af178fe2275029c5)

Comment 4.4

Line 359. m3/day

Response 4.4

m3/day is now revised to m3/day.

Comment 4.5

Following question 4.12 in first version. Suggest to add how the bovine factor was used to correct the gene copy of SARS-CoV-2 in Methods.

Response 4.5

The details of the calculation of the BCoV factor were provided in the previous publication of the group (https://doi.org/10.1016/j.scitotenv.2020.142939). Therefore, the following sentence is added included in order to prevent repetition.

“The calculation and explanation of how BCoV was used as an external reference are provided by Jafferali et al. [10].”
